# School-Based TGfU Volleyball Intervention Improves Physical Fitness and Body Composition in Primary School Students: A Cluster-Randomized Trial

**DOI:** 10.3390/healthcare11111600

**Published:** 2023-05-30

**Authors:** Darko Stojanović, Vladimir Momčilović, Marko Zadražnik, Igor Ilić, Admira Koničanin, Johnny Padulo, Luca Russo, Toplica Stojanović

**Affiliations:** 1Pedagogical Faculty in Vranje, University of Niš, 18000 Niš, Serbia; darkos@pfvr.ni.ac.rs (D.S.); vladmomcilovic@pfvr.ni.ac.rs (V.M.); 2Faculty of Sport, University of Ljubljana, 1000 Ljubljana, Slovenia; marko.zadraznik@fsp.uni-lj.si; 3Faculty of Sport and Physical Education, University of Priština—Kosovska Mitrovica, 38218 Leposavić, Serbia; igor.ilic@pr.ac.rs (I.I.); toplica.stojanovic@pr.ac.rs (T.S.); 4Department of Biomedical Science, Sports and Physical Education, State University of Novi Pazar, 36300 Novi Pazar, Serbia; akonicanin@np.ac.rs; 5Department of Biomedical Sciences for Health, Università degli Studi di Milano, 20133 Milan, Italy; johnny.padulo@unimi.it; 6Department of Human Sciences, Università Telematica Degli Studi IUL, 50122 Florence, Italy; 7Faculty of Physical Education and Sport, University of Banja Luka, 78101 Banja Luka, Bosnia and Herzegovina

**Keywords:** physical education, team sport, small-sided games, modified games, health-related fitness, adolescents

## Abstract

This study aimed to explore whether a 16-week Teaching Games for Understanding (TGfU) volleyball intervention could improve primary school students’ physical fitness and body composition. Eighty-eight primary school students (age 13.3 ± 0.3 years) were randomized to a TGfU volleyball intervention group (VG) or a control group (CG). The CG attended three regular physical education (PE) classes per week, while the VG attended two regular PE classes and a TGfU volleyball intervention that was implemented in the third PE class. Body composition components (body weight, body mass index, skinfold thickness, body fat percentage, and muscle mass percentage) and physical fitness (flexibility, vertical jumps (squat and countermovement jump—SJ/CMJ), 30 m sprint, agility, and cardiorespiratory fitness) assessments were performed pre-and post-intervention period. Significant interaction effects between VG and CG and pre- and post-test were found for the sum of five skinfolds (*p* < 0.0005, *ŋp*^2^ = 0.168), body fat % (*p* < 0.0005, *ŋp*^2^ = 0.200), muscle mass % (*p* < 0.0005, *ŋp*^2^ = 0.247), SJ (*p* = 0.002, *ŋp*^2^ = 0.103), CMJ (*p* = 0.001, *ŋp*^2^ = 0.120), 30 m sprint (*p* = 0.019, *ŋp*^2^ = 0.062), agility *T*-test (*p* < 0.0005, *ŋp*^2^ = 0.238), and VO_2max_ (*p* < 0.0005, *ŋp*^2^ = 0.253). Further examination revealed a greater improvement among VG students compared to CG students in certain body composition and physical fitness outcomes. Implementing a TGfU volleyball intervention in the physical education curriculum appears to have effective stimuli for reducing adiposity and promoting physical fitness levels in seventh-grade primary school students.

## 1. Introduction

Regular physical activity is recognized as beneficial for general health due to reducing the risk of non-communicable conditions such as obesity, diabetes, coronary heart disease, and stress [1]. The most recent World Health Organization recommendations on physical activity for children and adolescents emphasize the importance of engaging in moderate-to-vigorous physical activity (MVPA) for at least 60 min/day in order to provide health benefits [2]. However, there has been a prevalence of insufficient physical activity [3] and obesity [4] among adolescents over the last two decades. Regarding the serious detrimental implications of excess adiposity, the rise of obesity in childhood is a public health concern [5,6]. Students should spend no less than 50% of time participating in MVPA during physical education (PE) classes, as recommended by the U.S. Department of Health and Human Services [7]. While objectively measured MVPA during PE classes for Serbian students has not been studied to a certain extent, it is worth noting that PE classes in both European countries and the US frequently fail to achieve adequate MVPA levels among students [8,9,10]. Reviews that summarized studies on objectively measured physical activity in primary school PE classes reported that children spent a mean 32.6% [10] and 36.8% [8] of PE class time participating in MVPA. Besides physical activity, physical fitness is also recognized as an important determinant of children’s health, wellbeing [11,12], and academic achievement [13]. However, during the past two decades, there has been a significant decline in physical fitness levels among children and adolescents [14]. As a remedy to this issue, it is argued that physical education’s curricular aims, which are oriented towards an individual’s education and all-round development, have the potential to promote optimal growth, health-related physical fitness development, and psycho-social qualities through suitable physical activity levels [15]. Additionally, physical education classes offer a unique environment where children can benefit in domains such as affective and cognitive development. Engaging in a variety of physical activities, including sports, allows students to learn valuable social skills, such as participation, cooperation, competition, tolerance, and responsibility, while also promoting sportsmanship, fostering teamwork, and encouraging a lifelong commitment to an active and healthy lifestyle [16].

Nevertheless, it appears that the PE curriculum does not have sufficient intensity and stimuli to promote health-related fitness levels in children [17]. As a result of these findings, a substantial number of studies are suggesting various school-based PE interventions given the efficacy of these interventions in elevating the intensity of PE classes [18,19], physical activity levels of students [20], physical fitness [21,22,23,24,25], and students’ body composition [21].

In contrast to traditional technique drills and the full-court gameplay approach in physical education, Bunker and Thorpe [26] developed the “Teaching Games for Understanding” (TGfU) approach, which integrates skill execution, decision making, and tactical drills in modified small-sided games to ensure that all students can be actively involved in gameplay, and, therefore, possibilities for inactivity are reduced [20]. This approach has been widely recognized and applied across a variety of sports and physical activities including territorial games, soccer, basketball, handball, volleyball, floorball, baseball, and badminton [27]. Specifically, in volleyball, the small-sided volleyball game format is aimed at simplifying the game rules in particular settings where each player has the opportunity to interact with the ball more frequently. This is accomplished through reducing the number of players and the volleyball court area, which leads to greater gameplay intensity in a more favorable environment for learning basic volleyball techniques and tactics [28]. Furthermore, a previous study advocated that implementing the TGfU model of mini-volleyball into physical education classes promotes high levels of enjoyment and volleyball-specific skills enhancements [29].

However, recent literature revealed a research gap concerning the implementation of school-based volleyball interventions in physical education classes (following the TGfU approach) and their impact on health-related measures, including body composition and physical fitness. For this reason, the present study aimed to evaluate the effects of the TGfU volleyball intervention on body composition and physical fitness among 13-year-old primary school students. The implementation of the TGfU volleyball intervention in physical education was hypothesized to improve body composition and physical fitness outcomes more than the traditional physical education approach among 13-year-old primary school students.

## 2. Materials and Methods

### 2.1. Design and Participants

This study was a school-based, parallel-group, cluster-randomized controlled trial that recruited primary school students. A total of eighty-eight seventh-grade students (age 13.3 ± 0.3 years) from one primary school and six classes were included in this study. Randomization was performed at the class level. The students from six classes were randomly allocated to either a school-based TGfU volleyball training group (VG, three classes, n = 39 (20 boys and 19 girls)) or a control group (CG, three classes, n = 49 (25 boys and 24 girls)). Detailed sample characteristics are presented in Table 1. Students were enrolled if they fulfilled the following requirements: (a) they were in seventh grade and aged 13; (b) were not exempt from participating in physical education classes; (c) were not engaged in organized sports activities during the intervention period; (d) did not have any health issues (i.e., pediatric disease, orthopedic conditions, injuries, respiratory or cardiovascular disease); (e) voluntary participation in the study, which included a testing protocol for CG and testing protocol and exercise program for VG; and (f) they had a parent’s or legal guardian’s signed consent. Consent was also obtained from the physical education teachers and the school principal. Only students who participated in pre-and post-tests and had at least 85% intervention class attendance were considered for final processing. The student selection process is presented in a flow diagram (Figure 1).

### 2.2. Procedures

The study was conducted in the second school semester over 18 weeks between February and June 2018 (Figure 2). At the beginning of the second semester, two familiarization sessions were conducted to introduce all participants to the assessment protocol and the VG group to the structure, aims, and objectives of the volleyball training intervention. The first week was intended for baseline assessment of students’ anthropometry, physical fitness, and body composition in two alternate days during physical education classes (48 h elapsed time between the testing days). The requirements for participants were to maintain a normal diet, not eat food and drink a beverage at least two hours before the test, and not to participate in vigorous physical activity before the evaluation process. The first test day included sit and reach, squat jump, countermovement jump, agility T-test, and a 30 m sprint test. On the second test day, an assessment was conducted regarding body composition and cardiorespiratory fitness (20-m multistage shuttle run). Prior to the beginning of the testing protocol, the participants completed a standard warm-up (~10 min), comprising dynamic stretches, jogging, and specific drills incorporating test-related movements. Finally, after the 16-week intervention period, the same protocol was repeated for post-intervention assessment in the last week of the semester. All testing procedures were performed in the same order, in the school gym, in the morning hours (10 a.m.), and with the same trained personnel.

### 2.3. Intervention Program

According to the national curriculum, children in Serbia must attend two regular physical education classes and one “Physical Education—Selected Sport” class per week (2 + 1). The national curriculum for regular PE includes volleyball activities planned for the second semester of seventh grade, which cover technical drills and full-court play. Additionally, the PE—Selected Sport curriculum provides an opportunity for seventh graders to choose from different sports. However, in this study, all CG participants selected volleyball and followed national curriculum for volleyball sport sessions as detailed in Table 2. The CG students conducted regular PE classes and PE—Selected Sport (volleyball) classes (2 + 1 per week or 32 + 16 in total, respectively).

In contrast, besides regular PE classes twice a week, the VG students followed up the 16-week TGfU volleyball intervention (Table 2) with a frequency of one 45 min session per week (total number of sessions: 32 regular PE + 16 TGfU volleyball intervention). Each TGfU session was introduced with a three-part structure (introduction, main, and cool-down). The introduction part involved 10 min of moderate-intensity warm-up activities such as jogging, and dynamic stretching exercises (5 min), and volleyball-specific warm-ups such as skipping, hopping, jumping, and shuttle competitions with or without a ball in order to prepare the students for the following tasks. The main part of the session involved mini-volleyball and small-sided games (SSGs) in the format of 2 vs. 2, 3 vs. 3, and 4 vs. 4 (25–30 min). The last part of the session was the cool-down period (5 min). Due to the lack of equipment to assess the heart rate, we could not monitor the intensity during the exercise. Previous studies have shown that this game format provokes children to engage more frequently and with an intensity of 70–90% of HR_max_. [30,31,32]. The main part of each volleyball intervention session was based on the TGfU model, which emphasizes learning skills and tactical fundamentals of the volleyball game throughout modified games and allows participants to have contacts with the ball more frequently. The mini-volleyball and SSGs were played on multiple smaller courts in the sports gym with various sizes ranging from 4.5 to 6 m in width, 9 to 12 m in length, and divided by a net of 1.9 to 2 m in height. The most common SSGs were adapted from volleyball training and modified to accommodate the needs and abilities of the students. The teacher manipulated the rules during mini-volleyball and SSGs to exaggerate tactical issues or maintain or increase game intensity. For example, if the ball went out of play, the teacher maintained game flow and intensity by feeding in another ball instantly. The teacher could also increase the game intensity by adding an additional ball into the game or altering the playing area. At the end of the intervention, VG students played full-court competitive volleyball (6 vs. 6) following official volleyball rules.

In summary, the main difference between VG and CG was content covered in third class, as presented in Table 2.

### 2.4. Measures

#### 2.4.1. Body Composition

Measurements of skinfold thicknesses were obtained at five skinfold sites (calf, suprailiac, subscapular, triceps, biceps) using a calibrated (0.2 mm) caliper (GPM, Switzerland) and following the guidelines suggested by Eston and Reilly [33]. The sum of skinfolds was calculated by summing the thicknesses of all five skinfold sites. Body weight, BMI, body fat, and skeletal muscle mass percentage were measured with a bioelectrical impedance body composition monitor BF511 (Omron, Japan). Before body composition measurements, participants were requested to follow procedures, as outlined by Kyle et al. [34].

#### 2.4.2. Physical Fitness

Sit and Reach. The test for flexibility assessment was adopted from the Eurofit test battery [35]. The flexibility of the lower back and hamstrings was measured in a seated position with the feet flat against a box and the fingertips on the edge of the top plate. The participant attempted to reach as far forward as possible along the measuring scale by flexing trunk and hip joints while maintaining straight knees and fully extended arms. The maximum reach of the two trials was recorded to the nearest centimeter [36]. The reliability of Sit and Reach test was satisfactory with intraclass correlation coefficient (ICC) value of 0.94.

Vertical jump performance. The participants performed the squat jump (SJ) and countermovement jump (CMJ) protocols described by Bosco et al. [37]. They were given the instruction to place and maintain their hands on hips in order to minimize the influence of arm movement to the height of their vertical jump [38]. The SJ was performed in a 90-degree knee flexion position with a 3 s pause before the jump, whereas the CMJ was performed in an upright position before the countermovement phase. Each participant performed two maximal SJs and CMJs, with approximately two minutes of rest between jumps. Vertical jump heights were measured using Optojump (Microgate, Italy) with precision of 0.1 cm. The reliability of SJ and CMJ measurements was satisfactory, with ICC values of 0.89 and 0.88, respectively.

Sprint. The participants were instructed to perform two 30 m sprints with maximal effort in the school gym with 2–3 min of active recovery between sprints. Sprint times were recorded using Witty (Microgate, Italy), a timing system with photocells which were placed approximately 0.75 m above the ground. To prevent the timing gate from being triggered too early, the starting line was placed 0.5 m in front of the first timing gate. The participants undertook to sprint in standing start position when they were ready. The best of two times was recorded within 0.01 s accuracy. The 30 sprint measurement reliability was satisfactory, with ICC value of 0.90.

Agility. Following the procedure described by Semenick [39], agility was evaluated utilizing an agility T-test. A Witty photocell gate (Microgate, Italy) was placed approximately 0.75 m above the ground at the starting line. The participants’ times were recorded within 0.01 s accuracy. To prevent the timing gate from being triggered too early, the starting line was placed 0.5 m in front of the first timing gate. The participants undertook to sprint in standing start position when they were ready. Pauole et al. [40] demonstrated satisfactory validity and reliability of the agility T-test.

Cardiorespiratory Fitness. The 20 m shuttle-run test was adopted from the Eurofit test battery. Throughout this test, participants ran between two lines 20 m apart. An audio tape provided the running pace. The initial velocity of 8.0 km·h^−1^ was increased by 0.5 km·h^−1^ each minute from beginning of the test. The test was finished for the participants who could not follow the pace (i.e., unable to reach the three-meter zone located in front of each 2 m line at the moment of audio signal two times consecutively) [41]. Ramsbottom et al. [42] provided equation for estimating maximal oxygen uptake (VO_2max_).

### 2.5. Statistical Analysis

Descriptive data for all variables were reported as means and standard deviations. The Shapiro–Wilk test was used to check the normality assumption in pre- and post-intervention results. A 2 × 2 repeated measures ANOVA was performed to examine main effects for pre- vs. post-intervention (time) and VG vs. CG (group), as well as interactions for time × group on the body composition and physical fitness outcomes. An analysis of covariance (ANCOVA) was performed to further examine the effects, accounting for pre-test differences. The partial eta squared (*ŋp*^2^) was used for effect size interpretation and was categorized as small (0.01), medium (0.06), and large (0.14), following Cohen’s benchmarks [43]. Regarding the within-group changes, effect size was interpreted as Cohen’s *d* and categorized as trivial (*d* < 0.2), small (*d* > 0.2–0.6), moderate (*d* > 0.6–1.2), large (*d* > 1.2–2.0), and very large (*d* > 2.0–4.0) [44]. The statistical significance level was set at *p* ≤ 0.05. Data analyses were conducted using SPSS version 23 (IBM, Armonk, NJ, USA).

## 3. Results

Utilizing the two-way repeated measures ANOVA, we conducted an analysis to explore the main effects of each factor (time and group), along with their interaction effects. This assessment provided valuable insights into the effects of the proposed volleyball intervention in comparison to the regular physical education curriculum, specifically in relation to body composition and physical fitness outcomes.

However, due to the significant differences observed between the groups during pre-test measurements (Table 3), we conducted an ANCOVA to obtain additional insights and clarify the effects of the intervention. By including the pre-test measurements as a covariate in the analysis, the ANCOVA accounted for the influence of the initial differences between the groups, allowing for a more accurate assessment of the intervention’s effect.

### 3.1. Body Composition

The results of the two-way repeated measures ANOVA for body composition outcomes are presented in Table 3. The results showed significant group by time interaction effects for the majority of the body composition outcomes, with the exception of weight and BMI (*p* > 0.05). Simple main effects for significant interactions are presented in Figure 3. Significant within-group (VG) changes were found in weight, sum of skinfolds, body fat percentage and muscle mass percentage, with small to moderate effect sizes of 0.50, 0.67, 0.62, and 0.65, respectively. Moreover, significant within-group (CG) changes were found in weight, sum of skinfolds, body fat percentage, and muscle mass percentage, with small to moderate effect sizes of 0.71, 0.32, 0.47, and 0.46, respectively. Over the intervention period, VG students increased muscle mass and decreased skinfold thickness and body fat, while CG students showed the opposite trend. The ANCOVA results revealed that after adjusting for the pre-test differences, the VG students exhibited significantly greater improvements in the sum of skinfolds (*p* = 0.002, *ŋp*^2^ = 0.116), body fat percentage (*p* = 0.000, *ŋp*^2^ = 0.151), and muscle mass percentage (*p* = 0.000, *ŋp*^2^ = 0.224) compared to the CG.

### 3.2. Physical Fitness

The results of the two-way repeated measures ANOVA for physical fitness outcomes are presented in Table 3. The results showed significant group by time interaction effects for all physical fitness components, with the exception of sit and reach (flexibility) (*p* > 0.05). Simple main effects for significant interactions are presented in Figure 4 and Figure 5. Significant within-group (VG) changes were found in sit and reach, squat jump, countermovement jump, 30 m sprint, agility *T*-test and VO_2max_, with small to moderate effect sizes of 0.57, 0.76, 0.86, 0.63, 1.10, and 0.92, respectively. Moreover, a significant within-group (CG) change was found in sit and reach, with a small effect size of 0.34. VG students increased in vertical jump, sprinting, and agility performance, and cardiorespiratory fitness, while CG students exhibited no significant changes except reduced flexibility over the intervention period. The ANCOVA results revealed that after adjusting for the pre-test differences, the VG students exhibited significantly greater improvements in the sit and reach (*p* = 0.007, *ŋp*^2^ = 0.088), squat jump (*p* = 0.010, *ŋp*^2^ = 0.080), countermovement jump (*p* = 0.002, *ŋp*^2^ = 0.112), agility *T*-test (*p* = 0.000, *ŋp*^2^ = 0.209), and VO_2max_ (*p* = 0.000, *ŋp*^2^ = 0.275) compared to the CG. 

## 4. Discussion

The main purpose of this study was to determine the effects of a 16-week TGfU volleyball intervention on changes in the body composition indicators and physical fitness levels of seventh-grade primary school students. The main findings showed that the VG students who participated in the TGfU volleyball intervention for 16 weeks exhibited a decreased subcutaneous fat thickness and total body fat percentage, as well as an increased muscle mass percentage. Moreover, all physical fitness outcomes, except flexibility, were improved among VG students over the intervention period. In contrast, the results demonstrated that children who participated in the 16-week normal PE curriculum had higher total body fat percentage, lower muscle mass percentage, and lower cardiorespiratory fitness capacity compared to baseline scores for the CG. The remaining physical fitness outcomes did not alter in the CG. The obtained results in the CG were expected, given that previous research has identified that the traditional PE curricula has insufficient class intensity and time to provide significant changes in health-related outcomes [9,45,46]. Similar to our findings, CG students (regular PE classes) exhibited an increased body fat percentage over 11 weeks according to Carrasco et al. [47]. The evidence above suggests that existing PE programs are not adequate to address the demands of students, particularly in adolescence, when participation in MVPA is crucial for promoting optimal physical fitness and reducing excess adiposity. A study by Wang and Wang [20] revealed that the mean MVPA time during a TGfU basketball intervention was considerably longer compared to the traditional technique-based group among secondary school students. However, in the case of young soccer athletes, the TGfU approach had a less pronounced impact on the duration of physical activity. Specifically, only light physical activity duration showed favorable results for the intervention group [48]. Therefore, it is suggested that school-based interventions based on the TGfU approach could be employed to promote PA and reach the proposed minimum of 50% of MVPA time during PE classes.

The findings that were obtained in the present study regarding the effects of the intervention on body composition were consistent with the results of earlier research that evaluated school-based interventions either within the PE curriculum or throughout after-school interventions. To our knowledge, there are only two studies that have investigated the effects of small-sided volleyball games programs on body composition among adolescents [49,50]. In our previous study, a similar skill-based and SSG volleyball intervention as part of the PE curriculum demonstrated a reduction in adiposity and an increase in muscle mass in both genders [49]. Furthermore, the 12-week volleyball SSG intervention decreased body fat percentage (4.3%), but without a significant difference compared to a control group of overweight adolescent girls [50]. Small-sided volleyball games can simulate the physiological demands of competition with an increase in heart rate (HR) above 85% HR_max_ during exercise [30]. A study that investigated a soccer-specific SSG intervention revealed that a game format with fewer participants (3 vs. 3) increased the number of ball interactions and mean HR values (>70% HR_max_) [51]. Furthermore, soccer-specific SSGs resulted in a reduction of total body fat in boys (7%) [47], and boys and girls (3.7%) [21] after 11-week interventions. Several interventions with a mean HR_max_ greater than 70% showed reduced skinfold thickness [52,53] and body fat [54], and increased muscle mass [52,53,54] among adolescent populations. Improvement in the VG with regards to body composition markers may be due to the TGfU volleyball intervention, but it must be considered that the seventh-graders were undergoing rapid growth onset in adolescence [55]. A growth spurt in puberty is characterized by rapid changes in body composition, particularly in body fat and muscle mass tissue [56,57,58]. Therefore, it is essential to closely monitor certain biological variables that influence the aforementioned changes during puberty onset in order to appropriately design PE programs that accommodate the needs of children and their developmental patterns.

In line with previous research, the 16-week TGfU volleyball intervention was significantly more effective than the traditional PE curriculum in improving students’ levels of physical fitness. In several studies, the authors incorporated similar concepts from various team sports [20,21,22,23] and individual sports [59], and found positive impacts on physical fitness.

In the present study, the results from vertical jump performance variables in the VG showed significant improvement in SJ and CMJ after the intervention of 9.1% and 9.9%, respectively. Similar findings were achieved following an eight-month after-school small-sided volleyball intervention, with a 3.0% improvement in CMJ [60]. Another study found that adolescent girls who participated in after-school small-sided multi-sport games showed a significant increase in their CMJ of 7.5% [61]. However, the results are not directly comparable since the authors proposed these interventions as additional after-school activities rather than interventions within PE; therefore, they should be interpreted with caution. The implementation of volleyball SSGs in previous studies showed considerable increases in vertical jumps in 15-year-old volleyball players [30] as well as standing long jump performance in 12-year-old students [62]. Moreover, tactical volleyball drills (2 vs. 2, 3 vs. 3, and 4 vs. 4) and mini-volleyball introduced as intervention additional to regular PE classes, promoted positive changes in horizontal jumping abilities among eleven-year-old boys [63]. According to supportive findings, children aged from 10 to 12 who trained in mini-volleyball showed significant improvements in their vertical jump in comparison to their peers [64].

The VG students significantly improved speed and agility by lowering times by 3.4% and 1.8%, respectively. These results are consistent with a recent meta-analysis that demonstrated moderate effects of SSGs on sprint performance and a large effect on agility [65]. Improvements in sprint and agility performance were expected and can be partially explained by the intervention program, which included warm-up exercises, such as short sprints, shuttle competitions, and particularly SSGs, and mini-volleyball in the main part of session. In fact, previous studies demonstrated that SSGs can contribute to more frequent agility maneuvers, short runs, and change-of-direction actions, which can improve sprint and agility performance [30,65,66].

The most recent evidence confirmed that cardiorespiratory fitness, as the most powerful marker of health in childhood and adolescence, has been in decline over the past decades [67] and can affect future cardiovascular risk factors [68]. Cardiorespiratory fitness among the VG students was significantly improved by 1.6% after a TGfU volleyball intervention. Cocca et al. [21] suggested that a TGfU model multi-sport intervention within PE can increase cardiorespiratory fitness in primary school students. Morra and Hansen (2012) suggested that the TGfU model can promote cardiorespiratory fitness, since they monitored students’ HR levels during TGfU classes and discovered that students remained within or exceeded their targeted HR zone for a prolonged time compared to conventional PE settings, as cited in Cocca et al. [22]. Another study, carried out by Petrusic et al. [61], demonstrated that after-school small-sided football, volleyball, handball, and basketball games significantly increased aerobic fitness (10.6%) in comparison to regular PE among adolescent girls.

Although the improvements in physical fitness levels among VG students can be attributed to the TGfU volleyball intervention as a more effective pedagogical approach compared to the conventional PE curriculum, it should be noted that the group of adolescents who participated in this study were in the middle of growth and development spurts, which are critical periods for certain physical fitness component improvements. Critical periods are characterized by an accelerated developmental phases, during which specific effects cause the body to react more strongly [55]. This indicates that an individual’s potential to successfully improve performance and certain skills is at its maximum during these critical periods. According to a recent meta-analysis [65], volleyball SSG training appears to be more beneficial for improving jumping and sprinting performance than SSGs adopted in soccer and handball, as well as agility performance in comparison to basketball SSGs. One cross-sectional study compared performance characteristics among adolescent boys practicing nine different sports (badminton, basketball, gymnastics, handball, judo, soccer, table tennis, triathlon, and volleyball), and the results showed that boys who practiced volleyball had the best results in vertical and horizontal jumps and the second-best results in 30 m sprinting [69]. Nonetheless, it is worth noting that TGfU appears to induce some additional benefits, which may contribute to the effectiveness of the conducted intervention as mediators. According to recent studies, students who participated in the TGfU volleyball intervention found themselves more motivated [70] and experienced more enjoyment [29] in contrast to their peers who attended conventional PE classes.

There were no injuries reported in the VG during the volleyball intervention. Volleyball is considered as one of the team sports with the lowest injury rate. Nonetheless, during the volleyball intervention, we adopted injury prevention strategies. In addition to their regular warm-up and stretching routines, the children were required to wear indoor sports shoes with good ankle support and grip and knee pads to absorb the impact from falling and diving on their knees. Additionally, we reduced the number of free balls on the floor and in the air during the volleyball drills.

Finally, the present study had some limitations. First, a between-gender comparison was not examined. Second, maturity status indicators, such as the Tanner stage and PHV (Peak Height Velocity), should be included in future studies. Third, there was a lack of physical activity monitoring during the intervention and monitoring of out-of-school activities. Fourth, dietary intake monitoring was not included. Fifth, because the intervention was carried out on seventh-grade students, the results should not be generalized for children or adolescents of different ages. However, apart from the few limitations of the present study, the strength of the implemented TGfU volleyball intervention is that it is efficient for improving health-related markers, such as body fat and cardiorespiratory fitness, which are strongly related to health risks.

## 5. Conclusions

Considering the results obtained during the present study, it can be concluded that the 16-week TGfU volleyball intervention, comprised of small-sided games and mini-volleyball, can promote positive changes in the majority of the health-related fitness components among 13-year-old primary school students. The proposed intervention was designed with the intent to provoke more frequent engagement in moderate-to-vigorous volleyball activities in order to reduce students’ adiposity and improve physical fitness. In terms of meeting students’ natural physical development demands, the proposed pedagogical approach in PE settings appears to be more effective than traditional techniques and full-court volleyball gameplay. Future studies should explore whether school-based interventions following the TGfU approach with small-sided and modified games from various sports impact health-related fitness among children across different age groups.

## Figures and Tables

**Figure 1 healthcare-11-01600-f001:**
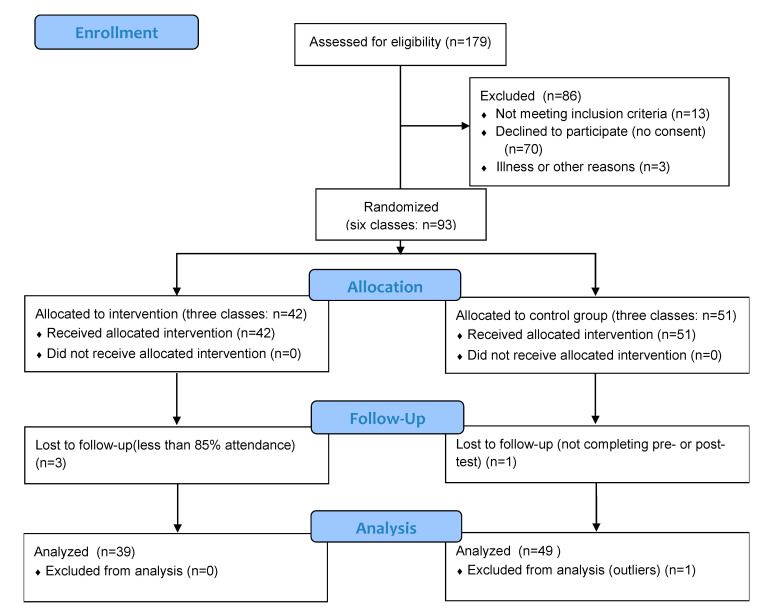
Flow chart diagram of subjects included in the study, randomization, and analysis.

**Figure 2 healthcare-11-01600-f002:**
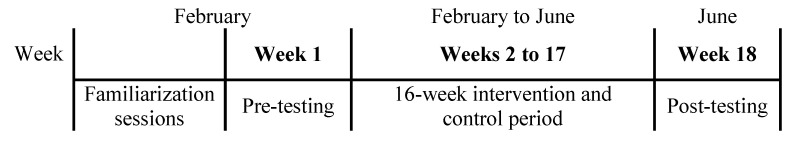
Study timeline.

**Figure 3 healthcare-11-01600-f003:**
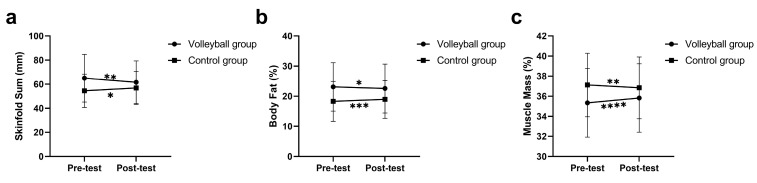
Significant interaction between group and time for sum of skinfolds (**a**), body fat % (**b**), and muscle mass % (**c**). Significant at * *p* < 0.05; ** *p* < 0.01; *** *p* < 0.001; **** *p* < 0.0001.

**Figure 4 healthcare-11-01600-f004:**
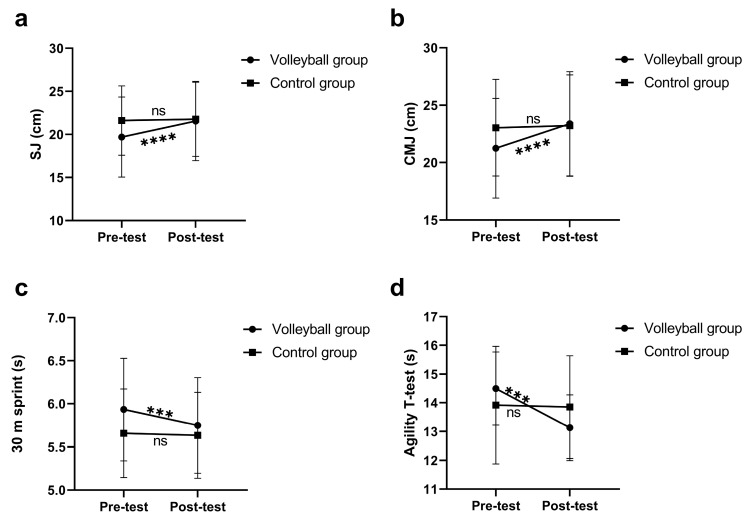
Significant interaction between group and time for SJ (**a**), CMJ (**b**), 30 m sprint (**c**), and agility *T*-test (**d**). Significant at *** *p* < 0.001; **** *p* < 0.0001.

**Figure 5 healthcare-11-01600-f005:**
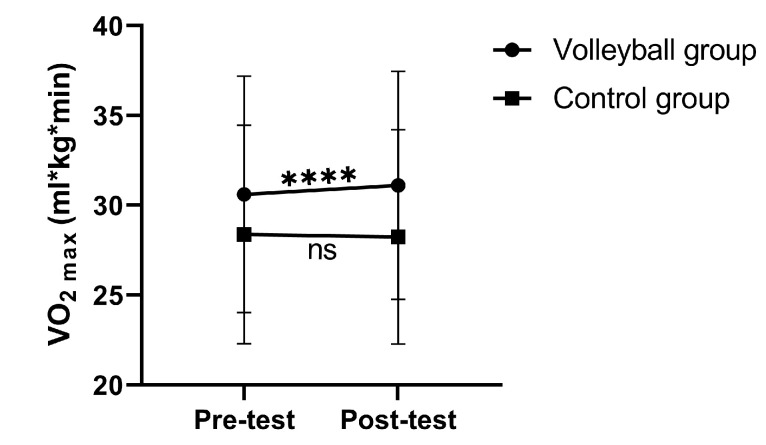
Significant interaction between group and time VO_2max_. Significant at **** *p* < 0.0001.

**Table 1 healthcare-11-01600-t001:** Anthropometric characteristics before and after TGfU intervention.

Variables		Volleyball Group		Control Group	
		Pre-Test	Post-Test	Pre-Test	Post-Test
Body Height (cm)	M	164.3 ± 7.4	166.6 ± 7.6	166.3 ± 9.9	168.8 ± 10.1
	F	160.7 ± 6.0	162.3 ± 6.1	159.1 ± 5.9	160.5 ± 6.0
	Total	162.6 ± 6.9	164.5 ± 7.2	162.8 ± 8.9	164.7 ± 9.2
Body Weight (kg)	M	56.4 ± 10.1	57.7 ± 11.4	56.8 ± 10.8	57.8 ± 11.6
	F	55.3 ± 12.4	55.8 ± 11.8	48.0 ± 7.2	49.4 ± 7.3
	Total	55.9 ± 11.2	56.8 ± 11.5	52.5 ± 10.1	53.7 ± 10.5
BMI (kg·m^−2^)	M	20.8 ± 2.9	20.7 ± 3.3	20.4 ± 2.6	20.2 ± 2.9
	F	21.3 ± 4.2	21.1 ± 3.9	18.9 ± 2.1	19.1 ± 2.2
	Total	21.1 ± 3.6	20.9 ± 3.6	19.7 ± 2.5	19.6 ± 2.6

Values are presented as mean ± standard deviation; M, male students; F, female students; BMI, body mass index.

**Table 2 healthcare-11-01600-t002:** Comparison between 16-week TGfU volleyball intervention and PE—Selected Sports class units and contents.

Session/ Week	VG; TGfU Volleyball Intervention		CG; PE—Selected Sports Class	
	Unit	Content	Unit	Content
1	Movement with and without a ball	Footwork drills with the low–mid–high stance	Technical drills	Forearm pass, overhand receive
2	SSG	Various tactical drills 2 vs. 2		Passing and receiving in pairs
3	Mini-Volleyball	3 vs. 3 with overhand pass only		Passing and receiving in pairs
4	SSG	Various tactical drills 2 vs. 2		Overhand pass and spike parallel
5	Mini-Volleyball	3 vs. 3 with forearm pass only		Overhand pass and spike diagonal
6	Volleyball-specific skills—jumping	One- or two-footed jumps, line hops, jacks, rope jumps		Overhand pass and spike parallel and diagonal
7	SSG	Various tactical drills 3 vs. 3		Overhand pass and spike parallel and diagonal
8	Mini-Volleyball	3 vs. 3 with overhand and forearm pass only		Overhand pass and spike parallel and diagonal
9	SSG	Various tactical drills 3 vs. 3		Blocking
10	Mini-Volleyball	3 vs. 3 with overhand, forearm pass, and spike		Blocking
11	SSG	Various tactical drills 4 vs. 4		Blocking
12	Volleyball-specific skills—agility	Accelerations, shuttles, change of direction drills	Mini-Volleyball	2 vs. 2, 3 vs. 3
13	Mini-Volleyball	3 vs. 3 with overhand, forearm pass, spike, and block	Mini-Volleyball	2 vs. 2, 3 vs. 3
14	SSG	Various tactical drills 2 vs. 2 (9 × 3 m court size per side)	Full court competitive	6 vs. 6 with 6–0 rotation
15	Mini-Volleyball	4 vs. 4 with overhand, forearm pass, spike, block, and serve	Full court competitive	6 vs. 6 with 6–0 rotation
16	Full court competitive	6 vs. 6 with 4–2 rotation	Full court competitive	6 vs. 6 with 4–2 rotation

VG, volleyball group; CG, control group; TGfU, teaching games for understanding; PE, physical education; SSG, small-sided games.

**Table 3 healthcare-11-01600-t003:** Body composition and physical fitness changes from pre- to post-intervention in VG and CG.

Variables		Pre-Test	Post-Test	%	2 × 2 ANOVA	ES (*ŋp*^2^)
Body Composition						
Weight (kg)	VG	55.9 ± 11.2	56.8 ± 11.5	+1.61	group: NS	
	CG	52.5 ± 10.1	53.7 ± 10.5	+2.29	time: F = 31.432; *p* < 0.0005	0.268
					group × time: NS	
BMI (kg·m^−2^)	VG	21.1 ± 3.6	20.9 ± 3.6	−0.95	group: F = 4.238; *p* = 0.043	0.047
	CG	19.7 ± 2.5 *	19.6 ± 2.6	−0.51	time: NS	
					group × time: NS	
Sum of skinfolds (mm)	VG	65.0 ± 19.8	61.6 ± 17.8	−5.23	group: F = 4.995; *p* = 0.028,	0.055
	CG	54.5 ± 13.9 *	56.9 ± 13.7	+4.40	time: NS	
					group × time: F = 17.384; *p* < 0.0005	0.168
Body Fat (%)	VG	23.1 ± 8.0	22.6 ± 8.1	−2.16	group: F = 7.372; *p* = 0.008	0.079
	CG	18.3 ± 6.6 *	19.0 ± 6.3	+3.82	time: NS	
					group × time: F = 21.509; *p* < 0.0005	0.200
Muscle Mass (%)	VG	35.3 ± 3.4	35.8 ± 3.4	+1.42	group: F = 4.056; *p* = 0.047	0.045
	CG	37.1 ± 3.2 *	36.8 ± 3.1	−0.81	time: NS	
					group × time: F = 28.244; *p* < 0.0005	0.247
Physical Fitness						
Sit and Reach (cm)	VG	19.6 ± 8.8	21.9 ± 8.3	+11.73	group: NS	
	CG	18.0 ± 7.3	18.8 ± 7.1	+4.44	time: F = 20.018; *p* < 0.0005	0.189
					group × time: NS	
SJ (cm)	VG	19.7 ± 4.7	21.5 ± 4.6	+9.14	group: NS	
	CG	21.6 ± 4.0 *	21.8 ± 4.3	+0.93	time: F = 13.827; *p* < 0.0005	0.139
					group × time: F = 9.838; *p* = 0.002	0.103
CMJ (cm)	VG	21.3 ± 4.3	23.4 ± 4.5	+9.86	group: NS	
	CG	23.0 ± 4.2	23.2 ± 4.4	+0.87	time: F = 16.731; *p* < 0.0005	0.163
					group × time: F = 11.735; *p* = 0.001	0.120
30 m Sprint (s)	VG	5.9 ± 0.6	5.7 ± 0.6	−3.39	group: NS	
	CG	5.7 ± 0.5 *	5.6 ± 0.5	−1.75	time: F = 9.573; *p* = 0.003	0.100
					group × time: F = 5.733; *p* = 0.019	0.062
Agility *T*-test (s)	VG	14.5 ± 1.3	13.1 ± 1.1	−9.65	group: NS	
	CG	13.9 ± 2.0	13.9 ± 1.8	−0.01	time: F = 22.199; *p* < 0.0005	0.276
					group × time: F = 26.927; *p* < 0.0005	0.238
VO_2max_ (mL·kg^−1^·min^−1^)	VG	30.6 ± 6.6	31.1 ± 6.3	+1.63	group: NS	
	CG	28.4 ± 6.1	28.2 ± 6.0	−0.70	time: F = 9.198; *p* = 0.003	0.097
					group × time: F = 29.081; *p* < 0.0005	0.253

Pre-test and post-test values are presented as mean ± standard deviation; VG, TGfU volleyball intervention group; CG, control group; BMI, body mass index; CMJ, countermovement jump; SJ, squat jump; *, significantly different from VG at pre-test; %, percentage of change from pre- to post-test mean values; NS, non-significant; *ŋp*^2^, partial eta squared; ES, effect size.

## Data Availability

Data are available on request.

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
