# Peer review of "School-Based TGfU Volleyball Intervention Improves Physical Fitness and Body Composition in Primary School Students: A Cluster-Randomized Trial"

_healthcare, 2023, doi:10.3390/healthcare11111600_

Round 1
Reviewer 1 Report
Congratulations to the authors of the work, I consider that it is quite well detailed and applied the statistical tests.
Only some small improvements such as expanding the introduction a bit, that is, the bases on which the work sits at a theoretical level.
Reduce the use of acronyms and abbreviations, because throughout the article there is an abuse of acronyms that are sometimes written and repeated in parentheses, the nomenclatures within the text such as % in the results section is more appropriate to write their full word.
To clarify the acronym TGfU, I don't know if I've gone through something in the article, but it hasn't been clear to me what these acronyms mean.
Finally, a recommendation that I make is to write a few lines when starting a section, an example of this is found in the results point that is stated and goes directly to the subsection, prior to that, some introductory lines of everything that is presented in it would be appropriate. that point.
The improvement issues presented are appreciations in relation to the presentation of the work, because at a general level it is a work with a good quality.
Author Response
Response to the Comments of Reviewer 1
Thank you for your kind words and positive feedback. We appreciate the detailed review of our manuscript and for providing some insightful suggestions to strengthen our manuscript. We feel we have sufficient responses to each of the major concerns listed in Comments, which are further detailed below, and we hope that they alleviate the concerns regarding the approaches adopted in our manuscript.
Only some small improvements such as expanding the introduction a bit, that is, the bases on which the work sits at a theoretical level.
Our response: In our revised manuscript, we have expanded introduction on the goals of physical education to include additional aspects beyond growth and health-related fitness (Lines 66-71). Additionally, we have provided a bit more rationale for the implementation of the Teaching Games for Understanding approach in physical education (Lines 82-85).
Reduce the use of acronyms and abbreviations, because throughout the article there is an abuse of acronyms that are sometimes written and repeated in parentheses, the nomenclatures within the text such as % in the results section is more appropriate to write their full word
Our response: We appreciate your valuable feedback. We acknowledge the concern regarding the excessive use of acronyms and abbreviations in the manuscript. In our revisions, we have made efforts to strike a balance between readability and avoiding excessive use of acronyms and abbreviations. We have retained certain abbreviations, particularly in sentences where the phrase "physical education" is frequently mentioned or phrases that are too long, such as "moderate-to-vigorous physical activity (MVPA)", or "small-sided games (SSGs)", to maintain conciseness and avoid redundancy. Additionally, we have reviewed and revised the nomenclatures within the results section, opting to use the full words instead of symbols like "%" to enhance comprehension for readers.
To clarify the acronym TGfU, I don't know if I've gone through something in the article, but it hasn't been clear to me what these acronyms mean.
Our response:, The full word meaning of TGfU acronym as “Teaching Games for Understanding” can be found in Introduction section, particulary in lines 79-80. We revised the abstract section to include the full word meaning of the TGfU in order to enhance clarity for readers.
Finally, a recommendation that I make is to write a few lines when starting a section, an example of this is found in the results point that is stated and goes directly to the subsection, prior to that, some introductory lines of everything that is presented in it would be appropriate. that point.
Our response: We appreciate your suggestion regarding providing introductory lines at the beginning of Results section. In response, we have revised the manuscript to include brief introductory lines at the start of Results section.
Reviewer 2 Report
The present study is well established and could serve as a reference for improving PE classes. In general, the authors present a well-conducted study with concordance between the design method, results, and conclusions. Although the manuscript is generally correct, there are several points that should be improved. For example: Abstract, line 22. The author should write (whiteout initial) TGfU, the reader did not know what is this. Abstract, line 22. The author should write (without initial) TGfU; the reader does not know what is this. Introduction: The authors' hypotheses are worthy of writing. Your manuscript does not contain this information; please provide it. As shown in Table 1, were there any significant differences between the groups before the intervention? Please informate whether or not there was. Line 153, which is one of the biggest limitations of the present study (does not measure intensity). Have the authors collected data on RPE after each session? Line 234: the reference values of Cohen's D are not in accordance with 0.2, 0.6, and 0.8. Could you explain this difference? Table 3. To improve table eligibility, only data on significant differences should be provided.Author Response
Response to the Comments of Reviewer 2
Thank you for your kind words and positive feedback. We appreciate the detailed review of our manuscript and for providing some insightful suggestions to strengthen our manuscript. We feel we have sufficient responses to each of the major concerns listed in Comments, which are further detailed below, and we hope that they alleviate the concerns regarding the approaches adopted in our manuscript.
Abstract, line 22. The author should write (whiteout initial) TGfU, the reader did not know what is this.
Our response: We revised the abstract section to include the full word meaning of the TGfU in order to enhance clarity for readers.
Introduction: The authors' hypotheses are worthy of writing. Your manuscript does not contain this information; please provide it.
Our response: We appreciate your suggestion regarding providing research hypotheses in Introduction section. In response, we have revised the Introduction in order to provide a hypothesis statement at the end of the section.
As shown in Table 1, were there any significant differences between the groups before the intervention? Please informate whether or not there was.
Our response: We have included the differences between the groups at the pre-test in Table 3 for further clarity.
Line 153, which is one of the biggest limitations of the present study (does not measure intensity). Have the authors collected data on RPE after each session?
Our response: Thank you for bringing up the concern regarding the measurement of intensity in our study. While we did not directly measure intensity during the intervention sessions, we relied on previous studies that have reported the intensity levels associated with small-sided volleyball games. These studies have indicated that this type of activity typically elicits intensity levels above 70% of maximum heart rate (HRmax). Furthermore, to gain some insight into the intensity of the sessions in our pilot testing phase, we conducted a volleyball session and manually checked the participants' heart rate by palpation at the radial artery. While we acknowledge that this method is not as precise as using more sophisticated tools, it provided us with some indication that the participants were within the range suggested by previous studies mentioned in our manuscript.
We recognize the limitation of not directly measuring intensity, which is also mentioned in the study limitations. In future research, we will consider incorporating more precise methods of measuring intensity, such as heart rate monitors or accelerometers, to obtain more accurate and comprehensive data.
Line 234: the reference values of Cohen's D are not in accordance with 0.2, 0.6, and 0.8. Could you explain this difference?
Our response: In our analysis, we employed the thresholds recommended by Hopkins (2009), which are widely accepted and utilized in the fields of sport medicine and exercise science. Reference: 10.1249/MSS.0b013e31818cb278
Table 3. To improve table eligibility, only data on significant differences should be provided.
Our response: We appreciate your suggestion to improve the table's readability by including only data on significant differences. In the revised version of the manuscript, we have made the necessary adjustments to present only the data on significant differences in Table 3.
Reviewer 3 Report
see file attached

Author Response
Response to the Comments of Reviewer 3
Thank you for your kind words and positive feedback. We appreciate the detailed review of our manuscript and for providing some insightful suggestions to strengthen our manuscript. We feel we have sufficient responses to each of the major concerns listed in Comments, which are further detailed below, and we hope that they alleviate the concerns regarding the approaches adopted in our manuscript.
1.The explanations of the goals of physical education are very much focused on health-related aspects. This is of course understandable in the context of the present study, yet more attention should be paid to the concept of multiperspectivity of physical education. Body awareness, social aspects, opportunities for expression through sport are just a few aspects that are at least as important as health and performance. Addressing multiple channels of the students also increases the chance of lifelong sports participation, which then also has healthpromoting aspects. Therefore, the goals of physical education should be defined more precisely.
Our response: In response to your suggestion, we have revised our Introduction section to provide a more comprehensive understanding of the goals of physical education, encompassing not only health and performance but also these additional aspects. (Lines 66-71).
- You use a range of diagnostic methods for body composition and physical fitness. Why did you choose exactly these tests? What is the reason, what comparable studies can you refer to that have done this in a similar way?
Our response: The majority of the selected tests for body composition and physical fitness are widely recognized and commonly used in the field of physical education, particularly as part of health-related fitness assessments for students (1,2). Notably, tests such as the 20 m-shuttle run and sit-and-reach are included in the Serbian national test battery, which adheres to the Eurofit test battery standards. However, it is important to mention that the Agility T-test, while not a component of health-related fitness, is a skill-related component of physical fitness. Although it may not be commonly used in school settings, it is the most frequently employed field sport-specific test, especially in sports like soccer, basketball and volleyball. Therefore, considering the specific context of our study and the need to assess sport-specific agility, we included the Agility T-test in our assessment battery.
(1) https://doi.org/10.3389/fped.2021.640028
(2) http://dx.doi.org/10.5281/zenodo.495725
- I am somewhat skeptical that the effects measured are actually due to the intervention alone. Have you looked more closely at the significant group x time interactions? The CG has significantly better values in many values in the baseline measurement. To control for your ANOVAS, have you also tried ANCOVAS and included the differences at baseline as a covariate? Are the differences e.g. in body weight, skinfolds, body fat and muscle mass at t1 significant?
Our response: We appreciate the reviewer's observation regarding the differences between the groups at baseline. As this study was based on cluster randomization, it was expected to have some variations between the groups at baseline. These differences are often inherent in cluster-randomized designs and can arise due to the nature of the randomization process.
In our analysis, we focused on examining the group x time interactions, which allow us to assess the differential changes over time between the intervention and control groups. While the baseline differences are noteworthy, our primary interest lies in evaluating the trajectory or slope of changes rather than the differences at baseline. We have performed the ANCOVA analysis as suggested, and the results for body composition outcomes (BMI, BF%, and MM%) remain significant.
For physical fitness outcomes, there were some changes in the significance levels. The sit-and-reach test showed a significant difference when baseline values were included as a covariate, while the 30 m sprint did not reach statistical significance. However, the remaining physical fitness measures continued to demonstrate significant differences between the intervention and control groups.
When examining the adjusted means, we observed that the intervention group (VG) consistently had better results in all outcome measures compared to the control group.
- The lack of control over sports activities outside the three school units is a major shortcoming. This makes the conclusions seem very speculative. Can you elaborate on why you did not include this in the survey? It is too late now to change this, but in the manuscript this omission should be more clearly presented and justified.
Our response: We appreciate the comment regarding the lack of control over sports activities outside of the three school units and its impact on the conclusions of our study. That brings us to the fact that we ommited to mention in the manuscript that one of the inclusion criteria for participants in our study was that they should not be engaged in organized sports activities during the intervention period. This criterion was implemented to minimize the potential confounding effects of external sports activities on the outcomes being measured. However, it is important to highlight that our study was designed as a parallel-group randomized controlled trial. The control group served as a comparison group that did not receive the TGfU volleyball intervention but continued with their regular physical education classes. This design allowed for the comparison of outcomes between the intervention and control groups, providing insights into the specific effects of the TGfU volleyball intervention while accounting for the influence of external factors. In the revised manuscript, we added a line regarding the mentioned inclusion criteria (lines 112-113).
Reviewer 4 Report
Comments to the Author
The authors of this article did an admirable job on an important topic, aimed to investigate whether a 16-week Teaching Games for Understanding (TGfU) volleyball intervention could improve the physical fitness and body composition of primary school students. This paper is well-organized, pertinent, and may add to the literature base of an important. However, there are several points that require further clarity;
1- Page 2, Lines 71-82: We know that TGfU is designed to help students and athletes develop a deeper understanding of game concepts and strategies through active and engaging learning experiences. However, you should in which sports and physical activities (including team sports such as soccer, basketball, or handball, as well as individual sports such as gymnastics, track and field, and martial arts, etc.) it is commonly applied. please add references
2- Page 2, Line 90: Please provide more detailed information about the students' sporting background (branch, sporting age, etc.).
3- Page 3, Line 107: In this table, you should show pre-post comparisons of men and women because the content of the current table is already presented in table 3. Please revise it.
4- Page 4, Lines 120-123: What is the elapsed time between the first and second test days? Also what time of the day the measurements took place? please provide details.
5- Page 4, Lines 152-153: Move this sentence to limitation (Due to the lack of equipment to..)
6- Page 4, Lines 153-154: What percent intensity did you train the children? and how did you determine and stabilize the exercise intensities throughout the follow-up? please explain.
7- Pages 7-8, Lines 251-268: It may be useful to give group x time interaction effect values inside the figures (for Figure 3, 4, and 5).
8- Page 9, Line 272: I noticed some inaccuracies in the percentage differences.! For example, in the Agility T-test VG group it should be 9.65% not -1.75%.! Similarly, please check and revise the others in detail.!
9- Page 10, Lines 328-329: Elaborate a little more on the work in this paragraph. I have also found some current studies for you, you can add them here or in your next paragraph.
https://doi.org/10.3390/ijerph19105816
https://doi.org/10.3390/ijerph17145008
GENERAL COMMENTS:
The topic is important but especially discussion sections should be improved significantly. Literature review is nonadequacy.

The language is clear, but perhaps could use a lit improvement.
Author Response
Response to the Comments of Reviewer 4
Thank you for your kind words and positive feedback. We appreciate the detailed review of our manuscript and for providing some insightful suggestions to strengthen our manuscript. We feel we have sufficient responses to each of the major concerns listed in Comments, which are further detailed below, and we hope that they alleviate the concerns regarding the approaches adopted in our manuscript.
1- Page 2, Lines 71-82: We know that TGfU is designed to help students and athletes develop a deeper understanding of game concepts and strategies through active and engaging learning experiences. However, you should in which sports and physical activities (including team sports such as soccer, basketball, or handball, as well as individual sports such as gymnastics, track and field, and martial arts, etc.) it is commonly applied. please add references.
Our response: We appreciate the reviewer's comment regarding the application of Teaching Games for Understanding (TGfU) in various sports and physical activities. To provide further support for the wide application of TGfU, we have incorporated reference to relevant study in the revised manuscript. (Lines 82-85).
2- Page 2, Line 90: Please provide more detailed information about the students' sporting background (branch, sporting age, etc.).
Our response: We acknowledge the importance of understanding the participants' prior sporting experiences in relation to their performance and response to the intervention. However, in our study, we did not collect detailed information about the students' sporting background, such as specific sports branches or sporting age.
One of the inclusion criteria for participants in our study was that they should not be engaged in organized sports activities during the intervention period. This criterion was implemented to minimize the potential confounding effects of external sports activities on the outcomes being measured.
3- Page 3, Line 107: In this table, you should show pre-post comparisons of men and women because the content of the current table is already presented in table 3. Please revise it.
Our response: We revised Table 1 as suggested.
4- Page 4, Lines 120-123: What is the elapsed time between the first and second test days? Also what time of the day the measurements took place? please provide details.
Our response: Page 4, Line 132 of the study states that the testing was conducted on two alternate days. However, to provide more precise information, we will specify that there was a 48-hour elapsed time between the testing days. Additionally, in Line 143, we mentioned that the measurements were taken in the morning hours, but to be more specific, we can specify that the measurements took place at 10 a.m.
5- Page 4, Lines 152-153: Move this sentence to limitation (Due to the lack of equipment to..)
Our response: In response to the comment, we would like to clarify that the limitation mentioned in Page 4, Lines 152-153 of the study has already been addressed in the study limitations section (Page 13, Lines 424-425).
6- Page 4, Lines 153-154: What percent intensity did you train the children? and how did you determine and stabilize the exercise intensities throughout the follow-up? please explain.
Our response: Thank you for bringing up the concern regarding the measurement of intensity in our study. While we did not directly measure intensity during the intervention sessions, we relied on previous studies that have reported the intensity levels associated with small-sided volleyball games. These studies have indicated that this type of activity typically elicits intensity levels above 70% of maximum heart rate (HRmax). Furthermore, to gain some insight into the intensity of the sessions in our pilot testing phase, we conducted a volleyball session and manually checked the participants' heart rate by palpation at the radial artery. While we acknowledge that this method is not as precise as using more sophisticated tools, it provided us with some indication that the participants were within the range suggested by previous studies mentioned in our manuscript.
We recognize the limitation of not directly measuring intensity, which is also mentioned in the study limitations. In future research, we will consider incorporating more precise methods of measuring intensity, such as heart rate monitors or accelerometers, to obtain more accurate and comprehensive data.
7- Pages 7-8, Lines 251-268: It may be useful to give group x time interaction effect values inside the figures (for Figure 3, 4, and 5).
Our response: While we appreciate your suggestion, we have decided not to migrate interaction effect values from Table 3 into the figures. As per the revisions made based on the second reviewer's feedback, Table 3 has been revised to enhance readability by removing non-significant results. Therefore, migrating only the interaction effects values into the figures would require addressing the presentation of the remaining values of main effects. We are open to considering new suggestions and exploring alternative ways to present the relevant information in a clear and concise manner.
8- Page 9, Line 272: I noticed some inaccuracies in the percentage differences.! For example, in the Agility T-test VG group it should be 9.65% not -1.75%.! Similarly, please check and revise the others in detail.
Our response: Thank you for your valuable input and keen observation. We genuinely appreciate your attention to detail and for bringing the inaccuracies in the percentage differences to our notice.
9- Page 10, Lines 328-329: Elaborate a little more on the work in this paragraph. I have also found some current studies for you, you can add them here or in your next paragraph
Our response: We are grateful for providing additional references to our manuscript. We have taken your suggestion into account and have included the two references you recommended in the revised version of the paper. (ref. no. 48 at Page 11, Lines 320-322 and ref. no. 59 at Page 11, Line 354).
Round 2
Reviewer 3 Report
Thank you for dealing with my remarks, the answers are all very comprehensible. Perhaps the explanations regarding the ANCOVA could also be included in the manuscript? This is not a must, but would prevent potential questions.
Author Response
Response to the Comments of Reviewer 3 (Round 2)
Thank you for dealing with my remarks, the answers are all very comprehensible. Perhaps the explanations regarding the ANCOVA could also be included in the manuscript? This is not a must, but would prevent potential questions.
Our response: Thank you for your suggestion. We have taken it into consideration and have made the necessary revisions in the manuscript. Additional explanation and results regarding ANCOVA can now be found in the revised version. Specifically, please refer to page 7, lines 246-248; page 8, lines 261-265, 277-280; and pages 7-8, lines 296-300.